# Artonin F Induces the Ubiquitin-Proteasomal Degradation of c-Met and Decreases Akt-mTOR Signaling

**DOI:** 10.3390/ph15050633

**Published:** 2022-05-21

**Authors:** Rapeepun Soonnarong, Ismail Dwi Putra, Nicharat Sriratanasak, Boonchoo Sritularak, Pithi Chanvorachote

**Affiliations:** 1Interdisciplinary Program of Pharmacology Graduate School, Chulalongkorn University, Bangkok 10330, Thailand; rapeepun.s@student.chula.ac.th; 2Center of Excellence in Cancer Cell and Molecular Biology, Faculty of Pharmaceutical Sciences, Chulalongkorn University, Bangkok 10330, Thailand; 6373013933@student.chula.ac.th (I.D.P.); nicharat.s@alumni.chula.ac.th (N.S.); 3Pharmaceutical Sciences and Technology Graduate Program, Faculty of Pharmaceutical Sciences, Chulalongkorn University, Bangkok 10330, Thailand; 4Departments of Pharmacology and Physiology, Faculty of Pharmaceutical Sciences, Bangkok 10330, Thailand; 5Department of Pharmacognosy and Pharmaceutical Botany, Faculty of Pharmaceutical Sciences, Chulalongkorn University, Bangkok 10330, Thailand; boonchoo.sr@chula.ac.th

**Keywords:** artonin F, apoptosis, lung cancer, c-Met, PI3K, Akt, mTOR, proteasomal degradation

## Abstract

Targeted therapies that selectively inhibit certain molecules in cancer cells have been considered promising for cancer treatment. In lung cancer, evidence has suggested that mesenchymal-epithelial transition factor (c-Met) oncoprotein drives cancer progression through its signaling transduction pathway. In this paper, we report the downregulation of c-Met by artonin F, a flavonoid isolated from *Artocarpus gomezianus*. Artonin F was found to be dominantly toxic to lung cancer cells by mediating apoptosis. With regard to its mechanism of action, artonin F downregulated c-Met expression, consequently suppressed the phosphatidylinositol-3 kinase/Akt/mammalian target of rapamycin signaling, increased Bax expression, decreased Bcl-2 expression, and activated caspase-3. The depletion of c-Met was mediated by ubiquitin-proteasomal degradation following co-treatment with artonin F, with the proteasome inhibitor MG132 reversing its c-Met-targeting effect. The immunoprecipitation analysis revealed that artonin F significantly promoted the formation of the c-Met–ubiquitin complex. Given that ubiquitin-specific protease 8 (USP8) prevents c-Met degradation by deubiquitination, we performed a preliminary in silico molecular docking and observed that artonin F blocked the catalytic site of USP8. In addition, artonin F interacted with the catalytic residues of palmitoylating enzymes. By acting as a competitive inhibitor, artonin F could reduce the degree of palmitoylation of c-Met, which affected its stability and activity. In conclusion, c-Met is critical for cancer cell survival and the failure of chemotherapeutic regimens. This novel information on the c-Met downregulating effect of artonin F will be beneficial for the development of efficient anticancer strategies or targeted therapies.

## 1. Introduction

Lung cancer is one of the most severe types of cancer and is the leading cause of cancer-related mortality worldwide. Lung cancer usually exhibits rapid growth and high resistance to chemotherapy [1]. The most significant obstacle to successful lung cancer management is therapeutic resistance and abnormal activation signaling. Several recent studies have focused on the development of new therapies that target specific signaling pathways in cancer cells, specifically molecules that target aberrant kinases [2]. Receptor tyrosine kinases (RTKs) are involved in various processes, including cell proliferation, survival, migration, invasion, and other cancer-related processes. Abnormalities in RTK activation have been linked to disease progression in various human cancers, making them promising drug targets for cancer treatment [3].

Mesenchymal–epithelial transition factor (c-Met) is an RTK, and it has generated increasing interest, with several clinical trials in progress [4]. For example, the overexpression of the *MET* gene has been linked to the development and poor prognosis of lung cancer [5]. In the majority of cancers, analyses have shown that c-Met expression has become amplified to levels higher than in normal cells. c-Met is also highly overexpressed in 60–80% of NSCLC. In addition, most of NSCLC cells express c-Met expression more than EGFR expression. The ligand for this receptor is hepatocyte growth factor (HGF). In normal cells, the binding of HGF to c-Met triggers a cascade of intracellular signals that mediate embryogenesis and wound healing [6]. Aberrant HGF/c-Met axis activation promotes tumor development and progression in cancer cells by activating the phosphatidylinositol-3 kinase (PI3K)/Akt, Ras/mitogen-activated protein kinase (MAPK) pathway and other signaling pathways [7]. A previous study showed that the c-Met receptor is linked to the development or progression of non-small cell lung cancer (NSCLC) when it is active for an extended period [8]. An in vitro and in vivo study showed that blocking of c-Met signaling can inhibit growth and cause the apoptosis of tumor cells, which is the main mechanism by which chemotherapeutic drugs kill cancer cells [9].

Protein palmitoylation is a lipid modification by modified cysteine thiols on proteins that produces a thioester with a palmitoyl group. Palmitoylation regulates the stability of membrane receptors by preventing their ubiquitination [10]. The palmitoylation of c-Met regulates its stability and trafficking in cancer cells [11]. Stabilized c-Met is bound to its ligand and activates downstream signaling pathways, including those of Akt, signal transducer and activator of transcription 3 (STAT3), and ERK, which leads to lung cancer cell growth, proliferation, and survival. The inhibition of palmitoylation can be a novel target for c-Met overexpression to prevent cancer invasion and metastasis [12]. As a result, targeting c-Met provides more opportunities for therapeutic approaches for lung cancer.

The mechanism of c-Met degradation has led to therapeutics targeting these pathways. The mechanism of c-Met degradation involves the ubiquitin-proteasome pathway. As the main mechanism, c-Met degradation is induced by the binding of HGF to c-Met, which leads to its dimerization and phosphorylation at Tyr1003, which is then labeled with ubiquitin for ubiquitination, resulting in the proteasome-activity-mediated lysosomal degradation of c-Met [13]. Therefore, the conjugation of ubiquitin to plasma membrane receptors is the major component of the regulatory mechanism for their internalization and lysosomal degradation. However, deubiquitination, the opposite process, is also critically involved in regulating the degradation of c-Met by removing monoubiqutin and polyubiquitin chains from ubiquitin-conjugated proteins, which results in the inhibition of protein degradation [14]. Therefore, a balance between ubiquitination and deubiquitination determines the outcome of internalized receptors and their downstream signaling. Ubiquitin-specific protease (USP) 8 is a deubiquitinating enzyme that plays an important role in enhancing cell proliferation and promoting cells to enter the S-phase during the cell cycle [15]. Deubiquitinating enzymes play a role in the endocytosis of membrane proteins from the cell surface and thus control lysosomal traffic/degradation. A previous study showed that USP8 regulates c-Met-mediated degradation in lung cancer. Given that USP8 cleaves the linkage between ubiquitin and c-Met, it inhibits c-Met ubiquitination and thus protects it from degradation [16]. For these reasons, the use of an USP8 inhibitor to overcome c-Met resistance in lung cancer can serve as a viable therapeutic option.

c-Met is an important target for cancer therapy and is found to be overexpressed in about 60–80% of NSCLC [17]. Several studies indicated that the flavonoid structure haa been described as a inhibitory effect of c-Met. Therefore, our compound artonin F could also be acts as the same property. Artonin F is a flavonoid lipid molecule isolated from the wall of the *Artocarpus gomezianus*. These plants contain a large diversity of bioactive compounds, including many prenyl flavonoids and cyclized derivatives that have revealed interesting anticancer properties [18,19]. The most important compound in the series is artonin E, the subgroup of compounds designated artonins, with the derivative artonin E as a lead anticancer agent [20]. Artonin E belongs to the class of organic compounds known as xanthone prenylated flavonoids [21]. Interestingly, artonin F also containing a xanthone functional group. The previous study showed that the xanthones are responsible for the cytotoxic to cancer cells [22]. The previous study of artonin E, which has a chemical structure similar to that of artonin F with slightly difference, demonstrated its efficacy as an anticancer compound in several cancer types [23,24]. In addition, a previous study of artonin E demonstrated its suppression of cancer cell motility through Akt signaling pathway [25]. For this reason, the investigation of the cytotoxic of artonin F could provide important information for the design of new anticancer agents.

## 2. Results

### 2.1. Artonin F Induces Apoptosis in Human Lung Cancer Cells

The cytotoxicity of artonin F was assessed to investigate its potential anticancer activity in human lung cancer cells. The 3-(4,5-dimethythiazol-2-yl)-2,5-diphenyl tetrazolium bromide assay was used to assess the effects of different concentrations of artonin F (0–50 μM) on various NSCLCs. The results revealed that artonin F showed significant cytotoxic effects at concentrations of 20–50 μM in A549 and H292 cells. Moreover, at concentrations of 40–50 μM, artonin F showed significant cytotoxic effects on H460 cells. As a result, artonin F significantly reduced viability in NSCLC cells in a concentration-dependent manner compared with that in untreated controls (Figure 1B–D). To determine the mode of cell death induced by artonin F, we identified the characterized apoptotic cells using a nuclear staining assay. After 24 h of treatment, morphological changes in the form of condensed and/or fragmented nuclei were observed, indicating that artonin F mediated apoptosis in lung cancer cells at concentrations ranging from 10 to 50 μM, with a small percentage of necrotic cells (Figure 1E–J). In addition, flow cytometry analysis with annexin-V-fluorescein isothiocyanate (FITC)/propidium iodide was performed to confirm the mode of cell death. The findings confirmed that 50 μM artonin F induced significant apoptosis in H460, H292, and A549 cells compared with that in untreated cells (Figure 1K–P), confirming apoptosis as the mechanism of artonin-F-induced cytotoxicity.

### 2.2. Artonin F Triggers Apoptosis Cascade through Mechanisms That Involve c-Met Downregulation

The activation of apoptotic pathways in tumor cells is an important strategy for cancer treatment. Many natural products that have been identified as potential sources of new anticancer drugs work by inducing apoptosis in cancer cells. To establish the mechanism associated with artonin-F-induced apoptosis, we examined the major regulators, such as caspase 3, anti-apoptotic protein Bcl-2, and pro-apoptotic protein Bax, in H460, H292, and A549 cells. The cells were treated in the same circumstances as the previous experiments. Compared with the untreated control, the results of the Western blot analysis revealed that artonin F significantly cleaved and activated caspase-3, upregulated Bax expression, and downregulated Bcl-2 expression (Figure 2A,B). Taken together, artonin F induced apoptosis in lung cancer cells by upregulating the expression of pro-apoptotic proteins, resulting in cell death.

Previous research discovered interactions between pro-survival and apoptotic signals, which alter the balance between survival and cell death [26]. Accordingly, we performed a Western blot analysis to investigate the underlying mechanisms of major pro-survival proteins, such as PI3K, p-PI3K, Akt, p-Akt, mammalian target of rapamycin (mTOR), and p-mTOR. Furthermore, we detected the level of expression of upstream signaling pathways, such as c-Met. Lung cancer cells were treated with 0–50 μM artonin F for 24 h, and the results showed that the ratios of PI3K, Akt, mTOR to GAPDH were reduced in cells treated with artonin F at the concentrations that induced apoptosis. Furthermore, after 24 h of exposure to 50 M artonin F, the level of c-Met was significantly reduced (Figure 2C,D). Thus, the mechanism of action of artonin F can reduce c-Met, as well as PI3K, Akt and mTOR.

We utilized caspase inhibitor (Z-VAD-FMK) to test the influence of caspase on the total form of proteins PI3K, Akt, and mTOR. The cells were treated with artonin F in the presence of Z-VAD-FMK and analyzed for PI3K, Akt, mTOR, c-Met, and caspase-3. The results show that the Z-VAD-FMK could inhibit the cleavage of caspase-3. While the downregulation of the total forms of PI3K, Akt, and mTOR induced by artonin F could be prevented by Z-VAD-FMK, the caspase inhibitor did not affect the c-Met reduction (Figure 2E,F). These results suggest that the PI3K, Akt, and mTOR proteins were reduced via caspase-dependent manner in artonin-F-treated cells.

To confirm the effect of artonin F in mediating c-Met depletion, we performed a time-dependent analysis to determine the protein levels at 6, 12, and 24 h after treatment. The treatement of the cells with 50 µM artonin could decrease c-Met level as early as 6 h (Figure 3A). Aftonin F at 20–50 µM could reduced c-Met expression at 12 h after treatment. To confirm, a Western blot analysis at the similar treatment conditions was performed and the results show that only c-Met, but not the total forms of PI3K, Akt, and mTOR, significantly decreased at 6–12 h (Figure 3B,C).

### 2.3. Artonin F Decreased the Levels of c-Met and p-PI3K

We further confirmed the results of previous experiments, that is, the underlying mechanisms of artonin-F-induced apoptosis in lung cancer cells, and found that c-Met downregulation contributed to artonin-F-induced cell death. Next, we investigated the effect of artonin F on c-Met and PI3K signaling by immunofluorescence staining. Figure 3A–F shows that artonin F significantly decreased the levels of c-Met in lung cancer cells. Similar results from the immunofluorescence analysis showed that artonin F significantly decreased the levels of p-PI3K (Figure 4G–L). These results indicate that artonin F decreased the levels of c-Met and p-PI3K, confirming the results of the previous experiment. These findings suggest that artonin F has a c-Met downregulating effect, and the mechanism of action may be linked to c-Met degradation.

### 2.4. Artonin F Decreases c-Met Levels through the Induction of c-Met Proteasomal Degradation

Through its signaling transduction pathways, c-Met promotes cancer progression in a variety of tumors. A previous study revealed that protein turnover is influenced by ubiquitin-proteasome degradation and that c-Met is also degraded by proteasomes [27]. Therefore, targeting the c-Met degradation pathways can lead to effective therapeutic strategies. Because an earlier study discovered that artonin F significantly reduces c-Met levels, we investigated its c-Met regulation mechanism further. The proteasome inhibitor MG132 was used to confirm that the destabilization of c-Met by artonin F occurred through this degradation mechanism. Moreover, we used co-immunoprecipitation to examine the premise of ubiquitin-mediated c-Met degradation and measured the level of the c-Met–ubiquitin complex (poly Ub-c-Met) after treatment with 50 µM artonin F. The resulting immune complexes were then analyzed for protein ubiquitination by Western blotting using an anti-ubiquitin antibody. The results in Figure 5A–F show that the addition of MG132 (10 µM) to the artonin F treatment enhanced the formation of the c-Met–ubiquitin complex (Ub-c-Met) compared to the non-treated controls, indicating that artonin F mediated c-Met stability via ubiquitin-proteasome degradation.

### 2.5. Computational Modeling Analysis of the Binding of Artonin F to c-Met

We aimed to create preliminary in silico models that can assess the molecular docking between c-Met (AlphaFold model, P08581) and artonin F to investigate the possible mechanism of downregulation of c-Met by artonin F. Given the inexistence of the complete structure of c-Met in the protein database, the c-Met model from AlphaFold [28], followed by some energy minimization steps, was used. Figure 6A shows the Ramachandran plot of minimized c-Met model. The model provided up to 96.1% residues in most favored and additionally allowed regions. Thus, it can be reliably used for docking. The docking results (Figure 6B,C) showed that artonin F does not interact with the Cys624 and Cys894 residues. These residues were reported to be the most important sites of c-Met palmitoylation [12]. According to Coleman (2016), c-Met palmitoylation is important for its trafficking and stability. Thus, blocking the palmitoylation site or inhibition of c-Met palmitoylation can be one of the strategies to destabilize c-Met. In addition, artonin F, which does not interact with Cys624 and Cys894 residues, exhibited an unfavorable donor–donor interaction with Ser615, Leu614, and Glu 855. Unfavorable interactions reduce the interaction stability of the ligand and protein. Thus, this interaction between artonin F and the observed site emphasizes that the destabilization of c-Met by artonin F does not occur through the direct blocking of the palmitoylation site. A docking study was conducted to further target the palmitoylating enzyme. Human DHHC20 palmitoyltransferase (PDB ID: 6BML) (Figure 6D,E) and acyl-protein thioesterase 1 (APT1, PDB ID: 6QGN) (Figure 6F,G) were used to investigate the possible mechanism of c-Met palmitoylation inhibition of artonin F. The docking studies of artonin F at the catalytic pocket of human DHHC20 palmitoyltransferase and APT1 showed that it binds several key residues in the catalytic process of the enzymes. Table 1 presents the docking results of artonin F with palmitoylating enzymes. According to the results, in binding the catalytic residues of palmitoylating enzymes, artonin F may acts as a competitive inhibitor and may be able to reduce the degree of palmitoylation in proteins, particularly c-Met. Based on the binding affinity, artonin F is more likely to inhibit APT1 than DHHC20 palmitoyltransferase, and the affinity of artonin F toward APT 1 is more negative than that toward DHHC20. In addition, artonin F exhibited a hydrogen bond interaction with the Ser119 residue of APT1, which is one of the catalytic triads of APT1, with a distance of 2.08 Å.

### 2.6. Computational Modeling Analysis of the Binding of Artonin F to USP8

Artonin F was also docked using AutoDock Vina [29] around the catalytic triad of USP8 (PDB ID: 3N3K). After the structure was cleaned and before docking, the USP8 PDB file was validated using the Ramachandran plot (Figure 7A). The plot indicated that 99% of the residues were in the most favored and additionally allowed region. Thus, the USP8 structure from the PDB did not need further preparation. The docking result showed that artonin F interacted with two of the three catalytic triads of USP8, which are represented in Figure 7B,C, namely, Asp1084 through hydrogen bonding and Pi−anion interaction and His1067 through the Pi−Pi stacked interaction. The data are provided in Table 2. Possibly, artonin F may able to be inhibit the activity of USP8 by blocking its catalytic sites (Cys786, His1067, and Asp1084) [30]. USP8 is one of the deubiquitinase enzymes of c-Met. By inhibiting the activity of USP8, c-Met is not deubiquitinated, resulting in its proteosomal degradation.

### 2.7. Computational Modelling Analysis of the Artonin F Interaction with c-Met Compared to Foretinib Interaction with c-Met

A preliminary analysis of the interaction between artonin F and c-Met at the ATP binding site was compared to the interaction between foretinib and c-Met in following study. The docking of artonin F at the ligand/ATP binding site of c-Met was carried out using the X-ray crystallography data of c-Met (PDB ID: 6sd9). The PDB file missed some important residues, therefore the missing residues were added and modelled prior to docking using Modeller. The redocking of natural ligand (foretinib) was carried out to verify the docking parameters. The redocking of foretinib returned the binding affinity of −11.3 kcal/mol and RMSD value of 1.11 Å, which indicates that the parameter set was suitable for docking at the specific site. The redocking result of foretinib also demonstrates the key interaction of c-Met and foretinib as an ATP-competitive inhibitor, i.e., the π interaction with Tyr1230, interaction with the hinge region (Met1160) and Asp1222, and the occupation of a lipophilic pocket (Figure 8A–C).

In the meantime, the docking of artonin F at the aforementioned site show slightly different results. The docking study gave the binding affinity of artonin F at ATP-binding site of c-Met approximately −8.0 kcal/mol, which is higher than the foretinib. The higher the binding affinity means the weaker the interaction. Furthermore, there was a donor–donor unfavorable interaction between artonin F and Met1160, which could weaken the interaction (Figure 8D,E). Therefore, the docking results suggest that, in comparison to foretinib, artonin F was unlikely to serve as a c-Met ATP-competitive inhibitor. However, artonin F has enough binding affinity to interact with the c-Met receptor.

## 3. Discussion

Multidrug resistance and side effects make lung cancer treatment challenging [31]. Cancer cells can make mistakes in terms of the apoptotic pathway and oncoprotein alteration [32]. As a result, targeting the apoptotic pathway and oncoprotein may result in effective therapeutic strategies. Natural product molecules are a promising treatment option for lung cancer, with few side effects and a high treatment success rate [33]. Various natural product molecules have been shown to be effective and useful in sensitizing conventional agents. In this study, we found for the first time that artonin F, a natural compound extracted from *A. gomezianus*, has anticancer activity and induces apoptotic cell death in human NSCLC cells.

Apoptosis causes cell shrinkage and nuclear condensation. This is followed by blebbing of the plasma membrane and nuclear fragmentation, resulting in apoptotic bodies that can be stained with annexin-V/FITC [34]. The results in Figure 1 suggest that the number of apoptotic cells increased in a concentration-dependent manner. The apoptosis pathway is regularly regulated by anticancer drugs via alterations in pro-apoptotic and anti-apoptotic proteins [35]. Many anticancer drug target caspase-3, which regulates cell death, and cleaved caspase-3 is associated with nucleosome fragmentation [36]. In addition, anti-apoptotic proteins, such as Bcl-2 and Bcl-xl, are highly expressed in lung cancer tissues. Therefore, we tracked the pro-apoptotic and anti-apoptotic markers and caspase. The results revealed that cleaved caspase-3 was upregulated in response to artonin F treatment, whereas Bcl-2 was downregulated in a concentration-dependent manner (Figure 2A,B).

*Met* is a proto-oncogene that encodes a RTK called c-Met for HGF [37]. When the HGF ligand binds to the c-Met receptor, it causes dimerization and autophosphorylation, which can then activate downstream target pathways, including PI3K, MAPK, and STAT [38]. c-Met is essential for cell proliferation, morphogenesis, and shielding from apoptosis. Aberrant Met expression and activation have been linked to cancer cell survival, proliferation, and invasiveness in many cancers, most notably NSCLC [6]. In addition, a previous study reported that phosphorylated c-Met inhibits apoptosis through the PI3K/Akt pathway [39]. In the present study, artonin F mediated apoptosis by the downregulation of c-Met and decreasing PI3K/Akt/mTOR signaling for suppressing cell survival and increasing the levels of pro-apoptotic proteins for the induction of the caspase cascade (Figure 2C,D). This result is consistent with the findings of a previous study, which found that imidazopyridine hydrazone derivatives significantly inhibited c-Met phosphorylation, inhibited cell growth in three-dimensional spheroid cultures, and induced apoptosis in AsPc-1 cells [40]. Furthermore, targeting c-Met with salvianolic acid A through the Akt/mTOR signaling pathway enhanced the sensitivity of lung cancer A549 cells to cisplatin [41]. Thus, c-Met and its related signaling pathways are important clinical targets. Furthermore, we concluded artonin F mediated c-Met downregulation and PI3K downstream signal transduction (Figure 4). In accordance with evidence supporting the use of c-Met targeted therapy, we observed that the level of c-Met was significantly reduced. Furthermore, artonin F can induce c-Met degradation via the ubiquitin-proteasomal pathway, and when the selective proteasome inhibitor MG132 was used, the level of c-Met in the artonin F-treated cells was significantly restored (Figure 5). The regulation and degradation of c-Met involve complex processes. With a better understanding of its degradation mechanisms, more opportunities to target these pathways therapeutically can be achieved.

Previous studies indicated that c-Met is palmitoylated, which facilitates its trafficking and stability [12]. The inhibition of palmitoylation reduces c-Met expression post-transcriptionally in several cancer cell lines. Therefore, we used preliminary in silico models to predict the mechanism of action of artonin F, which downregulated c-Met. In this study, we found that artonin F may act as a competitive inhibitor at the catalytic pocket of palmitoylating enzymes. Artonin F inhibited APT1 more effectively than DHHC20 palmitoyltransferase, with a binding affinity of −9.27 kcal/mol, and exhibited a hydrogen bond interaction with the Ser119 residue in the APT1 catalytic site (Figure 6). Therefore, the instability of c-Met following artonin F treatment was not caused by the direct inhibition of the palmitoylation site. The various mechanisms of c-Met degradation have led to therapeutics targeting these pathways. In addition, with the downregulation of deubiquitinating enzymes with specific targets of therapeutic importance, USP8 deubiquitinates c-Met and protects it from degradation [15]. The overexpression of USP8 in cancer cells is known to promote cell growth. Consistent with these data, we investigated the molecular docking between artonin F and USP8 and found that artonin F could inhibit the activity of USP8 by blocking its catalytic site (Figure 7). Therefore, c-Met is not deubiquitinated, resulting in the promotion of proteosomal degradation of c-Met. In addition, targeting intracellular c-Met with c-Met kinase inhibitors represents another approach for the HGF–c-Met signaling pathway deactivation. Foretinib is a multitargeted RTK inhibitor that targets receptors for Met and VEGF [42]. The results from the in silico experiment showed that artonin F was unlikely to act as a c-Met ATP-competitive inhibitor, compared to foretinib. Altogether, these results from in silico models could support that artonin F can inhibit palmitoyl transferase and USP8 that leads to the downregulation of c-Met activity and support protein degradation.

Artonin F exhibits the potential downregulation of c-Met by enhancing the ubiquitin-proteasomal degradation of c-Met. Finally, the results of the in silico models reported in this paper may be used to predict the mechanism of action of artonin F by inhibits USP8 catalysis and acting as a competitive inhibitor at the catalytic residues of palmitoylating enzymes. This compound may offer novel approaches for the improvement of c-Met-driven cancer.

## 4. Materials and Methods

### 4.1. Isolation of Artonin F

Dried and powdered root bark of *Artocarpus gomezianus* (5 kg) was macerated with methanol at room temperature, giving a methanol extract (485 g). The methanol extract was further triturated with ethyl acetate to yield ethyl acetate extract (96 g). This material was fractionated by vacuum liquid chromatography on silica gel (acetone-hexane, gradient) to give 5 fractions (A–E). Fraction B (2.6 g) was separated by column chromatography (silica gel, acetone-hexane, and gradient) and then subjected to repeat column chromatography (silica gel, acetone-dichloromethane, and gradient) to obtain artonin F (15 mg).

### 4.2. Cell Culture and Reagents

Human NSCLC cells, including H460, H292, and A549, were acquired from the American Type Culture Collection (Manassas, VA, USA). A549 was cultivated DMEM medium (Gibco, Grand Island, NY, USA), whereas H460 and H292 cells were cultured in RPMI 1640 medium (Gibco, Grand Island, NY, USA). The medium for cells cultured was supplemented with 10% fetal bovine serum (FBS) with 1% penicillin and streptomycin and 2 mM L-glutamine (Gibco, MD, USA). The cells were maintained in a 5% CO_2_ environment at 37 °C. Dimethyl sulfoxide (DMSO), propidium iodide (PI), Hoechst 33342, 3-(4,5-dimethylthiazol-2-yl)-2,5-diphenyltetrazoliumbromide (MTT), bovine serum albumin (BSA), Z-VAD-FMK, and antibody for ubiquitin (catalog no. u0508) were purchased from Sigma Chemical, Inc. (St. Louis, MO, USA). Phosphate buffer saline (PBS) and trypsin-EDTA were purchased from GIBCO (Grand Island, NY, USA). An apoptosis Kit (FITC) was purchased from ImmunoTools (Germany). Antibody for c-Met (#8198), p-PI3K (#4228), PI3K (#4292), p-Akt (#4060), Akt (#9272), mTOR (#2983), p-mTOR (#2971), Bcl-2 (#4223), Bax (#5023), caspase3 (#9662), and GAPDH (#5174) as well as peroxidase-conjugated secondary antibodies were obtained from Cell Signaling Technology, Inc. (Danvers, MA, USA).

### 4.3. Cytotoxicity Assay

Cells were seeded at the density of 1.5 × 10^4^ cells/well onto 96-well plates and incubated overnight. After that, cells were treated with various concentrations (0–50 μM) of artonin F for 24 h at 37 °C and analyzed for cell viability using MTT assay according to the manufacturer’s protocol (Sigma Chemical, St. Louis, MO, USA). In calculating the cell viability, the measured absorbance of treated cells was divided by the value of nontreated cells and reported as a percentage. All analyses were performed in 3 independent replicate cell cultures.

### 4.4. Nuclear Staining Assay

This method was applied to define apoptotic and necrotic cell death. H460, H292, and A549 cells seeded at the density of 1.5 × 10^4^ cells/well onto 96-well plates were incubated overnight. The cells were treated with artonin F at various concentrations (0–50 μM) for 24 h. at 37 °C. Afterward, the cells were incubated with 10 μg/mL of Hoechst 33342 and 5 μg/mL of propidium iodide (PI) for 30 min at 37 °C in the dark. These cells were visualized and imaged under a fluorescence microscope (Nikon ECLIPSE Ts2). The results are reported as the percentage of apoptotic cells and necrotic cells death.

### 4.5. Annexin V-FITC/PI Flow Cytometry

This method was applied to examine the stage of apoptosis and necrosis cells were determined with Annexin V-FITC Apoptosis Kit (ImmunoTools, Germany) by using flow cytometry. H460, H292, and A549 cells were seeded at a density of 1 × 10^5^ cells/mL in 24-well plates and incubated overnight. Then, cells were treated with artonin F at various concentrations (0–50 μM) for 24 h at 37 °C. Then, cells were detached from the well surface by using trypsin-EDTA (0.25%). Afterward, cells were suspended in 100 μL of 1× binding buffer and incubated in 5 μL of Annexin V-FITC and 1 μL of PI for 15 min at room temperature in the dark. Next, cells were analyzed by guava easyCyteTM flow cytometry systems (GuavaSoft Software version 3.3).

### 4.6. Western Blot Analysis

To determine protein expression for regulation in apoptosis and survival pathway, the treated cells were lysed into cellular lysates as previously described [43]. Equivalent amounts of protein from each sample were separated by SDS-PAGE and transferred to 0.2 µm polyvinylidene difluoride membranes (Bio-Rad, Hercules, CA, USA). The separating blots were blocked for 2 h with 5 % non-fat dry milk in TBST (Tris-buffer saline with 0.1% Tween containing 25 mM Tris-HCl (pH 7.5), 125 mM NaCl and 0.1% Tween 20) and incubated with specific primary antibodies against c-Met, p-PI3K, PI3K, p-Akt, Akt, mTOR, p-mTOR, Bcl-2, Bax, caspase3, and GAPDH at 4 °C overnight. Then, the membranes were washed in TBST and incubated with secondary antibodies for 2 h at room temperature. Finally, the protein bands were detected by using an enhancement chemiluminescence substrate (Supersignal West Pico; Pierce, Rockford, IL, USA) and exposed to the film.

### 4.7. Immunofluorescence

To determine the level of protein, c-Met, and p-PI3K, cells were seeded at the density of 1.5 × 10^5^ cells/well onto 96-well plates. After treatment cells with artonin F at 24 h., the cells were fixed with 4% (*w*/*v*) paraformaldehyde for 30 min. After that, cells were permeabilized with 0.1% (*v*/*v*) Triton-X for 20 min. Next, the cells were incubated with 3% (*w*/*v*) BSA for 30 min, washed with 1× PBS and incubated with an c-Met or p-PI3K antibody overnight at 4 °C, washed with 1× PBS and incubated with Alexa Flour 488 (Invitrogen, Waltham, MA, USA) conjugated goat anti-rabbit IgG (H + L) secondary antibody for 1 h at room temperature in the dark. Therefore, the cells were washed with PBS, co-stained with 10 μg/mL Hoechst 33342, and visualized and imaged using fluorescence microscopy (Nikon ECLIPSE Ts2), and the analysis was assessed by Image J software.

### 4.8. Co-Immunoprecipitation (Co-IP)

Cancer cell lines were pretreated with 10 µM MG132 for 30 min followed by 50 µM artonin F for 5 h; cells were washed with PBS and lysed with RIPA buffer, supplemented with protease inhibitor mixture and phosphatase inhibitor mixture at 4 °C for 20 min. The lysate was pre-cleared by incubating with beads at 4 °C for 1 h and then centrifuged at 12,000× *g* for 20 min at 4 °C. Cell supernatant was determined for protein content using the Bradford method. Cell lysates (500 μg of protein) were incubated with c-Met antibody at 4 °C overnight. After that, the lysate:immune complexes were incubated with protein G-conjugated agarose beads for 4 h at 4 °C. The immune:protein:bead complexes were washed five times with cold lysis buffer and suspended in 6× Laemmli sample buffer. The immunopurified samples are loaded on 7.5% acrylamide gels in order to probe for the presence of ubiquitinated c-Met by Western blot anti-Ub. The filters were incubated with anti-Ub antibody diluted 1:1000 in TBS-T 5% BSA for 1 h at room temperature. The protein complexes were analyzed for protein expression by immunoblotting assay.

### 4.9. Computational Method

Artonin F, as ligand, was prepared by drawing in 2D and 3D using Chem3D 19.0 (PerkinElmer Informatics, Waltham, MA, USA). The structure geometry was optimized in Gaussian 09W using Hartree-Fock 6-31G Basis Set. The target c-Met (UniProt P08581) structure was retrieved from AlphaFold prediction [28], while the human DHHC20 (PDB ID: 6BML) and acyl-protein transferase (APT1; PDB ID: 6QGN) was retrieved from rcsb.org. The predicted structure of c-Met was minimized first using YASARA force field minimization server [44]. The Ramachandran plot of the minimized c-Met was checked using PROCHECK server [45] and modelled prior to docking using Modeller [46]. Crystal structure of wild-type c-Met bound by inhibitor, highlighting the key role that residues Tyr1230 play in the inhibitor binding mode [47].

Protein–ligand in silico docking study was performed using AutoDock Vina [29]. Ligand and protein were prepared for docking using AutoDock Tools. Water and hetero-atoms from PDB file of proteins were deleted, polar hydrogen and Gasteiger charge of both protein and ligand were added, and the file was saved in pdbqt format. The grid box for docking was set at the centre_x1 = −30.95, centre_y1 = 15.02, centre_z1 = 4.24 and centre_x2 = −56.03, centre_y2 = −47.50, centre_z2 = 31.92 with dimension 20 × 20 × 20 Å for c-Met, centre_x = 25.40, centre_y = −51.20, centre_z = −55.20 with dimension 23 × 23 × 23 Å for DHHC20, and centre_x = −22.60, centre_y = −9.60, centre_z = −21.70 with dimension 23 × 23 × 23 Å for APT1 and centre_x = 15.02, centre_y = −18.94, centre_z = −4.63 with dimension 26 × 26 × 26 Å for USP8. The exhaustiveness was set at 100 and 10 ligand poses was produced. The best pose, which was characterized by having the lowest binding affinity, was analyzed further. Discovery Studio Visualizer was used to visualize the 2D and 3D interactions of protein and ligand (Discovery Studio Visualizer, version 2020).

### 4.10. Statistical Analysis

Data from three independent experiments are presented as mean ± standard error of mean (SEM). Multiple comparisons for statistically significant differences between multiple groups were performed using an analysis of variance (ANOVA), followed by Tukey’s post hoc test. *p*-value < 0.05 was considered as statistically significant.

## 5. Conclusions

This study examined the c-Met downregulating effect of artonin F in lung cancer cells. Artonin F induced a dramatic c-Met degradation through a ubiquitin-proteasomal mechanism and inhibited PI3K/AKT/mTOR signals. For mechanism, we proposed that artonin F interacted and inhibited deubiquitinating enzyme USP8, therefore enhancing the degradation. In addition, it also interacted with palmitoylating enzymes and destabilized c-Met (Figure 9). The promising anticancer activities of artonin F support its use in potential new approaches to treat c-Met-driven cancer.

## Figures and Tables

**Figure 1 pharmaceuticals-15-00633-f001:**
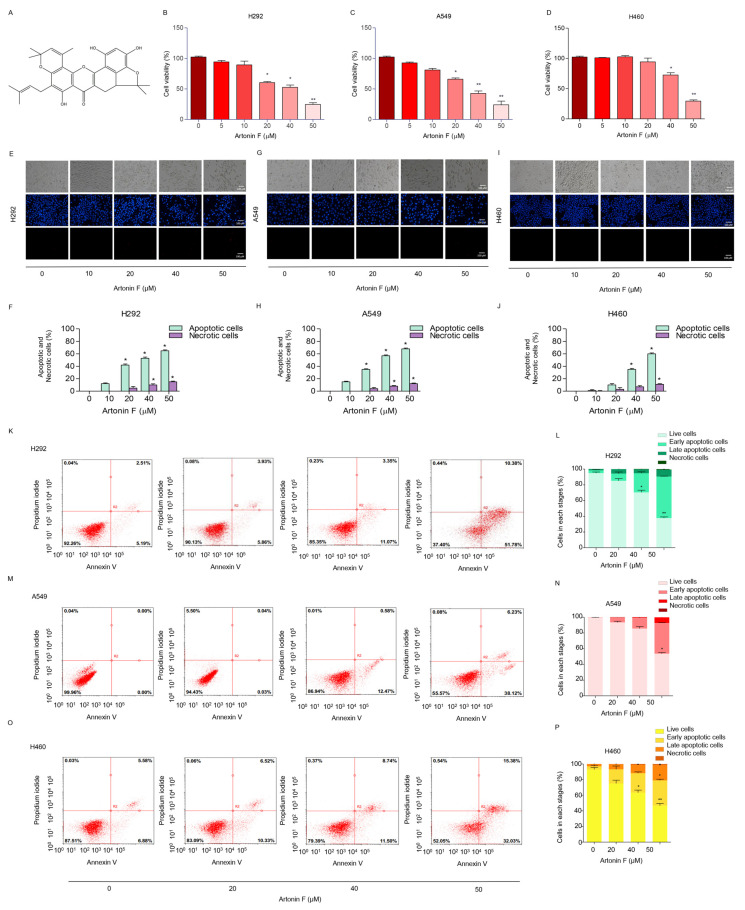
Cytotoxicity of artonin F on human lung cancer cells. (**A**) Chemical structure of artonin F. (**B**–**D**) Effect of artonin F on cell viability of lung cancer cells (H292, A549, and H460 cells) was assessed by the 3-[4,5-dimethylthiazol-2-yl]-2,5 diphenyl tetrazo-liumbromide (MTT) assay. (**E**–**J**) Morphology of apoptotic nuclei stained with Hoechst 33342 dye and propidium iodide in cells treated with artonin F, determined by visualized using fluorescence microscopy and percentages of nuclear fragmented and PI positive cells were calculated. (**K**–**P**) Apoptotic and necrotic cells were determined using Annexin V-FITC/PI staining with flow cytometry. Data are shown as the mean ± SEM (*n* = 3). * *p* < 0.05, ** *p* < 0.01 versus the non-treated control.

**Figure 2 pharmaceuticals-15-00633-f002:**
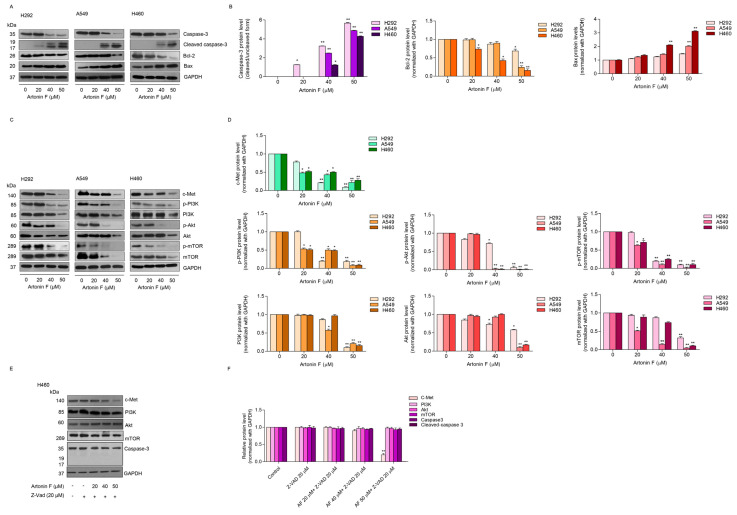
Artonin F triggers apoptosis through decreased c-Met and PI3K/Akt/mTOR signaling in lung cancer cells. (**A**,**B**) Effects of artonin F on the protein expression of the apoptosis-related proteins were detected by Western blot. Blots were reprobed with GAPDH to confirm equal loading of samples. The relative protein levels were calculated by densitometry. (**C**,**D**) Effect of artonin F on the protein expression of the c-Met/PI3K/Akt/mTOR marker was detected by Western blot. Blots were reprobed with GAPDH to confirm the equal loading of samples. (**E**,**F**) H460 cells were treated with artonin F and Z-VAD-FMK (20 µM) for 24 h. Expression of the proteins was detected by Western blot. Blots were reprobed with GAPDH to confirm the equal loading of samples. The relative protein levels were calculated by densitometry. Data are shown as the mean ± SEM (*n* = 3). * *p* < 0.05, ** *p* < 0.01 versus non-treated control.

**Figure 3 pharmaceuticals-15-00633-f003:**
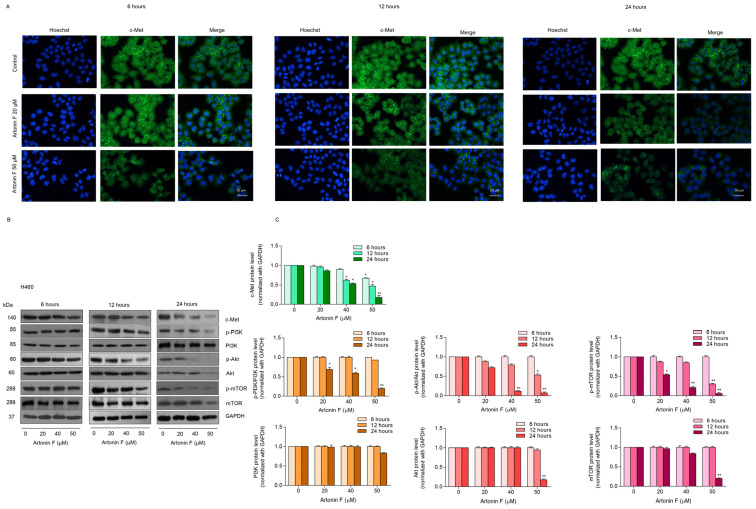
Artonin F causes a dramatic decrease in c-Met in the time-dependent analysis in the lung cancer cells. (**A**) Cells were stained with c-Met antibody (green fluorescence) and Hoechst 33342 (blue fluorescence). The expression of c-Met was determined by immunofluorescence. (**B**,**C**) The protein expressions were detected by Western blot. Blots were reprobed with GAPDH to confirm the equal loading of samples. The relative protein levels were calculated by densitometry. The relative protein levels were calculated by densitometry. Data are shown as the mean ± SEM (*n* = 3). * *p* < 0.05, ** *p* < 0.01 versus the non-treated control.

**Figure 4 pharmaceuticals-15-00633-f004:**
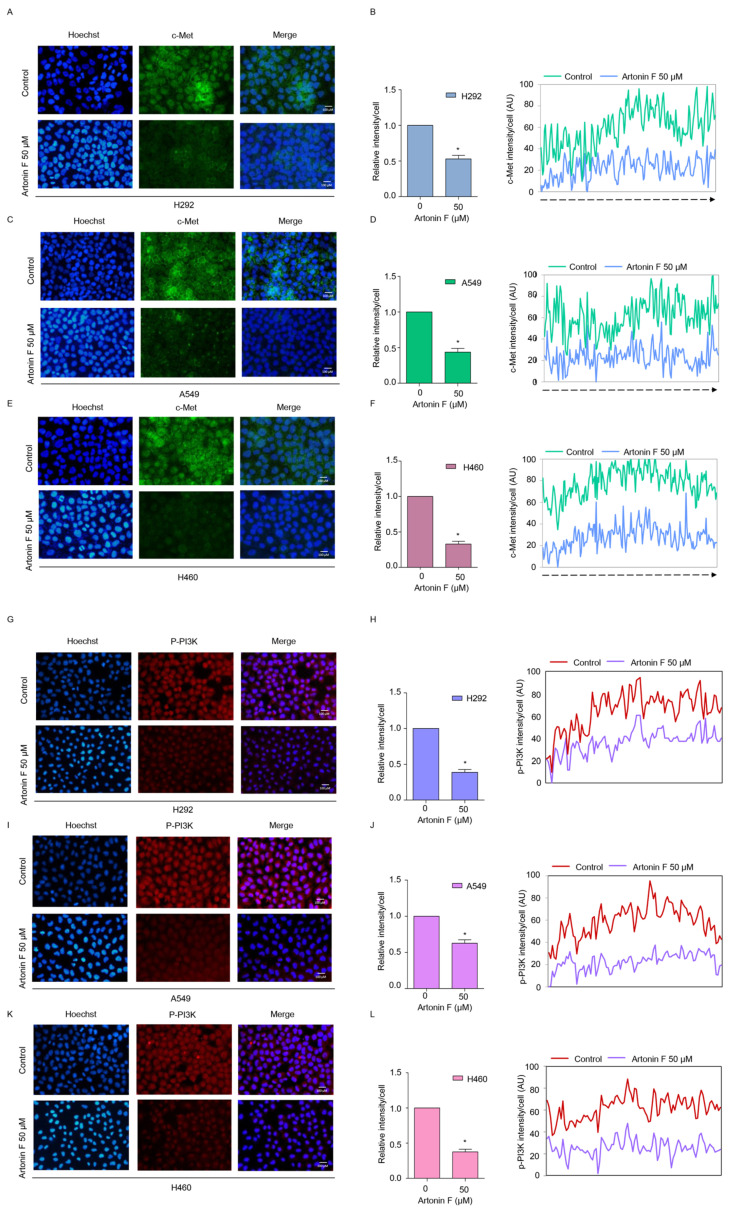
Artonin F decreases the levels of c-Met and p-PI3K. (**A**–**F**) The cells were stained with c-Met (green fluorescence) and Hoechst 33342 (blue fluorescence). The expression of c-Met was determined by immunofluorescence. (**G**–**L**) The cells were stained with p-PI3K (red fluorescence) and Hoechst 33342 (blue fluorescence). The expression of p-PI3K was examined using immunofluorescence. The fluorescence intensity was analyzed by ImageJ software. Data are shown as the mean ± SEM (*n* = 3). * *p* < 0.05, versus the non-treated control.

**Figure 5 pharmaceuticals-15-00633-f005:**
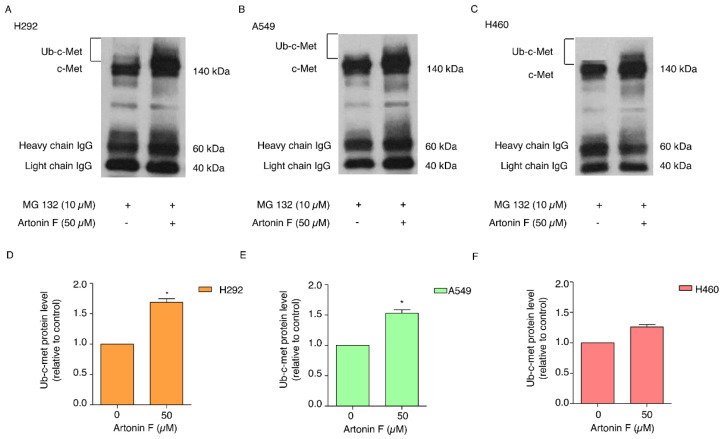
Artonin F induces ubiquitin-mediated c-Met proteasomal degradation. (**A**–**F**) Human cancer cells were pretreated with MG132 10 mM for 30 min, followed by treatment with artonin F 50 µM for 5 h. The protein lysates were collected and incubated with a mixture of beads and c-Met primary antibodies to pull out the protein of interest. Cell lysates are subjected to IP anti-c-Met and the immunoprecipitation complexes were analyzed for ubiquitin levels by Western blotting. The relative protein levels were calculated by densitometry. Data are shown as the mean ± SEM (*n* = 3). * *p* < 0.05 versus the non-treated control.

**Figure 6 pharmaceuticals-15-00633-f006:**
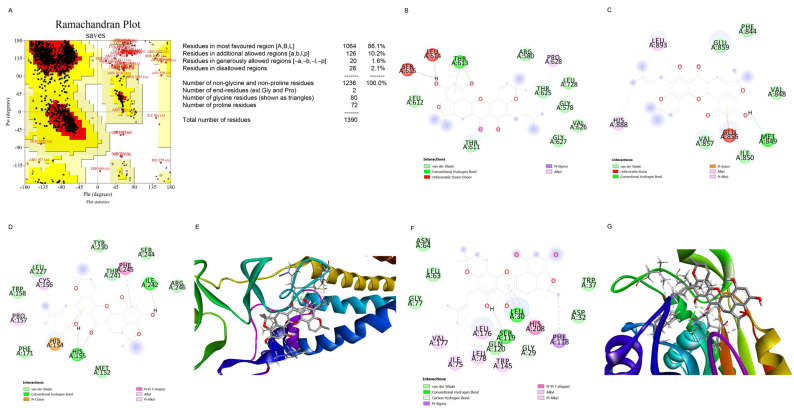
Molecular docking between c-Met and artonin F. (**A**) Ramachandran plot of c-Met model. (**B**) Interaction of c-Met and artonin F at near C624 residue. (**C**) Interaction of c-Met and artonin F at near C894 residue. (**D**,**E**) The 2D and 3D map interaction of artonin F with human DHHC20 palmitoyltransferase (PDB ID 6BML). (**F**,**G**) The 2D and 3D map interaction of artonin F with APT1 (PDB ID 6QGN).

**Figure 7 pharmaceuticals-15-00633-f007:**
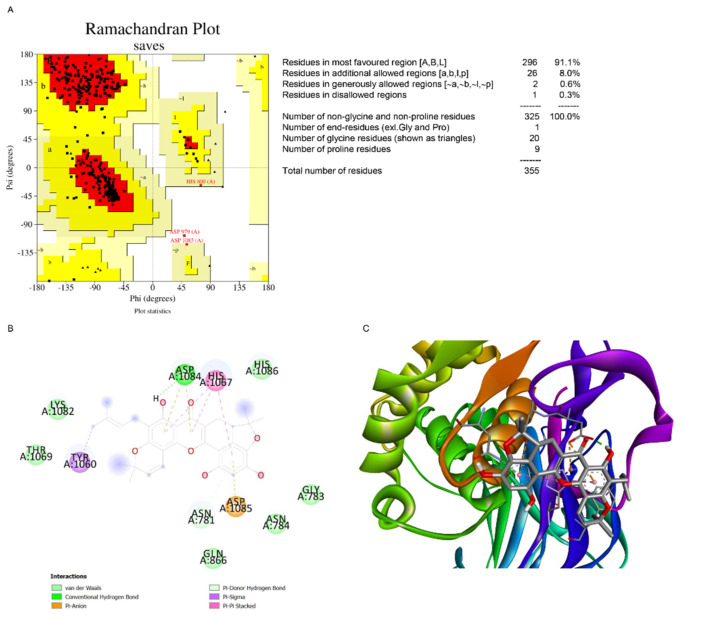
Molecular docking of artonin F at the catalytic site of USP8. (**A**) Ramachandran plot of USP8 (PDB ID 3N3K) generated from PROCHECK server. (**B**,**C**) The 2D and 3D interaction map of the docking result of artonin F at the catalytic site of USP8.

**Figure 8 pharmaceuticals-15-00633-f008:**
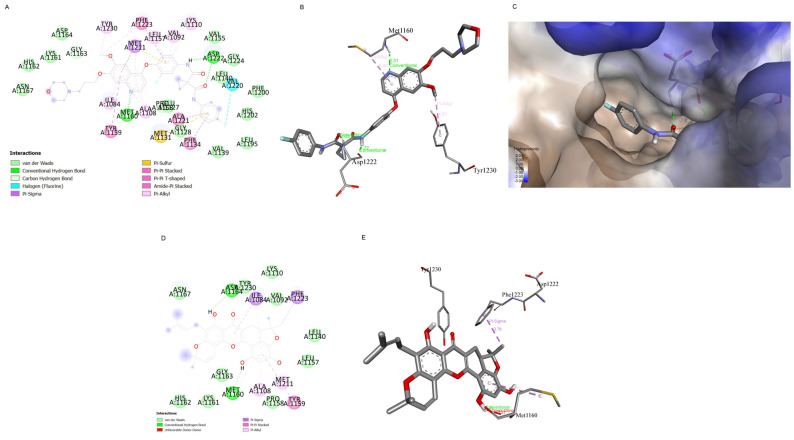
Molecular docking of artonin F interaction with c-Met compared with foretinib interaction with c-Met at the ATP-binding site. (**A**) The 2D interaction map of foretinib and c-Met (PDB ID: 6SD9). (**B**) The interaction of foretinib with the key residues in c-Met. (**C**) Lipophilic pocket occupation of terminal fluoro-phenyl group of foretinib. (**D**) The 2D interaction map of artonin F and c-Met (PDB ID: 6SD9). (**E**) The interaction of artonin F with the key residues in c-Met.

**Figure 9 pharmaceuticals-15-00633-f009:**
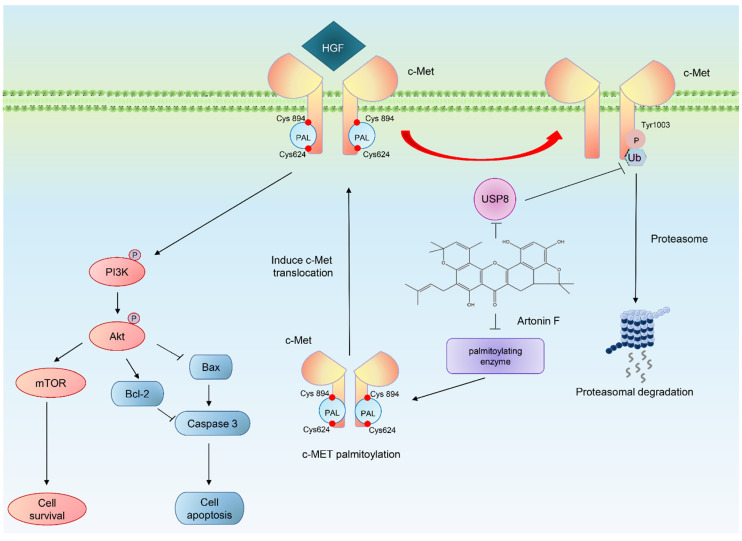
Schematic diagram of artonin F, which has a potential ability to induce apoptosis in human lung cancer cells and specifically triggers the ubiquitin-proteasome degradation of c-Met by inhibiting the activity of USP8 and at the catalytic residues of palmitoylating enzymes, this compound acts as a competitive inhibitor. The mechanism of action of artonin F is relatively specific through PI3K/Akt/mTOR signaling pathway.

**Table 1 pharmaceuticals-15-00633-t001:** AutoDock4 docking results of artonin F with palmitoylating enzyme.

Protein	Affinity (kcal/mol)	Estimated Ki (μM)	Type of Interaction	Interacting Residue(s)
hDHHC20 palmitoyl transferase	−8.03	1.30	Hydrogen bond	His155, Ile242
			Pi Cation	**His154**
			Pi–Pi T-Shaped	Phe245
			Alkyl and Pi–Alkyl	Pro157, **Cys156**, **His154**, Ile242
			Van der Waals	Trp158, Leu227, Tyr230, **Thr241**, Ser244, Arg246, Met152, Phe171
Acyl-protein thioesterase 1	−9.27	0.16	Hydrogen bond	Leu30, **Ser119**
			Pi–Sigma	Phe118, Leu176, Leu30
			Pi–Pi T-Shaped	**His208**
			Alkyl	Ile75, Val177, Leu78, Trp145, Leu176
			Van der Waals	Leu63, Asn64, Gly77, Gln120, Gly29, Trp37, Asp32, Ser209

Note: Bold residues are part of catalytic site.

**Table 2 pharmaceuticals-15-00633-t002:** Docking results of artonin F with USP8.

Binding Affinity (kcal/mol)	Interacting Residues
−6.8	Conventional Hydrogen Bond: Asp1084 [bond length = 1.93 Å] Pi interaction: Tyr1060, His1067, Asp1084, Asp1085

## Data Availability

Data are contained within the article.

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
