# Peer review of "Artonin F Induces the Ubiquitin-Proteasomal Degradation of c-Met and Decreases Akt-mTOR Signaling"

_pharmaceuticals, 2022, doi:10.3390/ph15050633_

Round 1

Reviewer 1 Report

The present study analyzed the anti-cancer effect of artonin F on lung cancer lines and its biological mechanism. Artonin F inhibited the cell survival by inducing apoptosis in those cancer cells. According to the further molecular biological analyses, the treatment with artonin F induced the degradation of c-Met by the ubiquitin-proteasome system. Furthermore, the computational modeling analysis clarified the interaction between artonin F and USP8. The reviewer considers the present study is well-constructed and has sufficient amount of experiments. The reviewer would like to ask some queries to the authors as described below.

  1. In the present manuscript, the authors indicated this study investigated the c-Met targeting treatment. For example, the authors described that the present study provides the novel information of c-Met targeting (in abstract) or that this compound may offer novel approaches for the improvement of cancers treatment targeting c-Met (in the last sentence of discussion). But the reviewer could not understand why the authors focused on c-Met as a target of artonin F. The reviewer would request to describe this reason in Introduction. Furthermore, the reviewer considers that the contribution of this study is not the c-Met targeting treatment because artonin F did not interact with c-Met molecule directly. In addition, the reference No.15 suggested that USP8 may altered the expression of not only c-Met but also other RTKs. So, the reviewer considers that this study may contribute to provide the novel method for targeting USP8. The reviewer would like to receive the authors’ opinion.
  2. The reviewer could not understand why the authors decided to investigate the effect of artonin F. Is there any prior research that shows the effectivity of artonin F?
  3. In Fig 2C, the authors showed the results of western blot to indicate the decreased expression of c-Met. But the reviewer considers the expression of other analyzed proteins (PI3K, Akt and m-TOR) also decreased. Do the authors consider that the decreased expression of those proteins is also derived from the inhibition of USP8 by artonin F?
  4. In Fig. 1D, the unit is absent.
  5. There are several typographical errors in the main text. The reviewer would request the proofreading.

Author Response

Response to reviewer comments

Reviewer: 1

The present study analyzed the anti-cancer effect of artonin F on lung cancer lines and its biological mechanism. Artonin F inhibited the cell survival by inducing apoptosis in those cancer cells. According to the further molecular biological analyses, the treatment with artonin F induced the degradation of c-Met by the ubiquitin-proteasome system. Furthermore, the computational modeling analysis clarified the interaction between artonin F and USP8. The reviewer considers the present study is well-constructed and has sufficient amount of experiments. The reviewer would like to ask some queries to the authors as described below.

  1. In the present manuscript, the authors indicated this study investigated the c-Met targeting treatment. For example, the authors described that the present study provides the novel information of c-Met targeting (in abstract) or that this compound may offer novel approaches for the improvement of cancers treatment targeting c-Met (in the last sentence of discussion). But the reviewer could not understand why the authors focused on c-Met as a target of artonin F. The reviewer would request to describe this reason in Introduction. Furthermore, the reviewer considers that the contribution of this study is not the c-Met targeting treatment because artonin F did not interact with c-Met molecule directly. In addition, the reference No.15 suggested that USP8 may altered the expression of not only c-Met but also other RTKs. So, the reviewer considers that this study may contribute to provide the novel method for targeting USP8. The reviewer would like to receive the authors’ opinion.

Response: We are grateful for the reviewer’s constructive comments, and we have carefully amended the manuscript accordingly. We have added the rationale of artonin F effect on the c-Met in the introduction part.

“c-Met is an important target for cancer therapy and is found to be overexpressed in about 60%–80% of NSCLC (1). Artonin F is a flavonoid lipid molecule isolated from the wall of the Artocarpus gomezianus. These   plants contain   a   large   diversity   of   bioactive compounds, including   many prenyl flavonoids   and cyclized   derivatives    which   have   revealed   interesting anticancer properties (2, 3). The most important compound in the series is artonin E, the subgroup of compounds designated artonins, with the derivative artonin E as a lead anticancer agent (4). Artonin E belongs to the class of organic compounds known as xanthone prenylated flavonoids (5). Interestingly, artonin F also containing a xanthone functional group. The previous study showed that the xanthones are responsible for the cytotoxic to cancer cells (6). The previous study of artonin E which has close chemical structure as artonin F demonstrated the efficacy as anticancer compound in several cancer types (7, 8). In addition, previous study of artonin E in suppression of cancer cell motility through Akt signaling pathway (9). For this reason, the investigation of the cytotoxic of artonin F could provide important information for the design of new anticancer agents.”

According to the reviewer’s suggestion, in the revision we have modified the description of “c-Met targeting” to “c-Met downregulating effect” in the abstract and conclusions part. In the discussion part we have modified the sentence “This compound may offer novel approaches for the improvement of cancers treatment targeting c-Met” to “This compound may offer novel approaches for the improvement of c-Met–driven cancer”.  We agree with the reviewer that this study is not the c-Met targeting treatment because artonin F did not interact with c-Met molecule directly. Artonin F can inhibit palmitoyl transferase and USP8 that leads to inhibition of c-Met activity and support protein degradation.

In addition, we performed extra-experimental in silico. The results showed that artonin F is likely to act as a c-Met ATP-competitive inhibitor when compared to foretinib (inhibitor targeting MET) the data is provided in result part section 2.7). Altogether, these results could support that artonin F can downregulated c-Met and operate as a c-Met competitive inhibitor.

References

  • Salgia R. Role of c-Met in cancer: emphasis on lung cancer. Seminars in oncology, (2009). 36, 52–58.
  • Prashanth J. , Suresh D., Potty V. H. and Sadananda Maiya P. In Vitro Anticancer and Hepatoprotective Activities of Artocarpus gomezianus. International Journal of Medical Sciences. (2014), 7, (18-23).
  • Losuwannarak, N.; Sritularak, B.; Chanvorachote, P. Cycloartobiloxanthone Induces Human Lung Cancer Cell Apoptosis via Mitochondria-dependent Apoptotic Pathway. In Vivo. (2018), 32, 71.
  • Bailly C. Anticancer mechanism of artonin E and related prenylated flavonoids from the medicinal plant Artocarpus elasticus. Asian Journal of Natural Product Biochemistry. (2021), 19, (44-56).
  • Yulin R., Esperanza J. Carcache de B., James R. F., Djaja D. S., Joanna E. B., Steven M. S., and Douglas A.K. Potential Anticancer Agents Characterized from Selected Tropical Plants. Journal of Natural Products. (2019), 82, (657-679).
  • Fábio F., Patrícia M. A. S., José X. S., Ana C. H., Daniela R. P., Carlos M. G., Carlos M. M., and Hassan B. A Pyranoxanthone as a Potent Antimitotic and Sensitizer of Cancer Cells to Low Doses of Paclitaxel. Molecules. (2020), 25, 1-17.
  • Yangnok K, Innajak S, Sawasjirakij R, Mahabusarakam W, Watanapokasin R. Effects of Artonin E on Cell Growth Inhibition and Apoptosis Induction in Colon Cancer LoVo and HCT116 Cells. Molecules. 2022, 7,2095.
  • Kanyaluck Y., Sukanda I., Ratchawin S., Wilawan M. and Ramida W. Effects of Artonin E on Cell Growth Inhibition and Apoptosis Induction in Colon Cancer LoVo and HCT116 Cells. Molecules. (2022), 7, 1-11.
  • Plaibua K., Pongrakhananon V., Chunhacha P., Sritularak B., and Chanvorachote P. Effects of Artonin E on Migration and Invasion Capabilities of Human Lung Cancer Cells. Anticancer Research. (2013), 33, 3079-3088.
  • Sierra JR, Tsao MS. c-MET as a potential therapeutic target and biomarker in cancer. Ther Adv Med Oncol. 2011 Nov;3(1 Suppl):S21-35.
  • Zanoaga O, Braicu C, Jurj A, Rusu A, Buiga R, Berindan-Neagoe I. Progress in Research on the Role of Flavonoids in Lung Cancer. Int J Mol Sci. 2019 Sep 2;20(17):4291.

  1. The reviewer could not understand why the authors decided to investigate the effect of artonin F. Is there any prior research that shows the effectivity of artonin F?

Response: We are grateful for the reviewer’s very insightful suggestions. The previous studies demonstrated the effect of Artonin E in suppressing cancer cell motility and promoting cytotoxicity through inhibition of Akt signaling pathway (1). Artonin E is an organic compound which contains xanthone group as a functional group and Artonin F has closely structure of this compound. Therefore, we consider that Artonin F may act as an anti-cancer agent same as the previous studied compound.

References

  • Plaibua K., Pongrakhananon V., Chunhacha P., Sritularak B., and Chanvorachote P. Effects of Artonin E on Migration and Invasion Capabilities of Human Lung Cancer Cells. Anticancer Research. (2013), 33, 3079-3088.

  1. In Fig 2C, the authors showed the results of western blot to indicate the decreased expression of c-Met. But the reviewer considers the expression of other analyzed proteins (PI3K, Akt and m-TOR) also decreased. Do the authors consider that the decreased expression of those proteins is also derived from the inhibition of USP8 by artonin F?

Response: We would like to thank the reviewer for raising this crucial point. Generally, Even USP8 plays an important role in protein degradation of several tyrosine kinase receptors. The reviewer is right about the possibility that the inhibition of USP8 may affect the total form of PI3K, Akt, and m-TOR. But the previous studies in NSCLC cells demonstrated that USP8 inhibitor could inhibit protein degradation of tyrosine kinase receptor but has no affect on the downstream signaling, such as Akt (1). In addition, the results of Fig 2A demonstrated that Artonin F can trigger caspase-3 activation in H292, A549, and H460 at concentration of 40-50 µM. In the most case, it is possible that the reduction of mentined proteins in Fig 2C may be the result of caspase clevage.

Moreover, the study of c-Met degradation, we performed the experiment at an early time.  The cells were treated with Artonin F at 50 µM for only 5 hours which may not yet trigger the caspase-3 activation, however, the result demonstrated celar induction of c-Met ubiquitination. Therefore, the degradation of c-Met in such case was not the consequence from the caspase-3 event.

Reference

  • Sanguine B., Sung-Young L., Jihoon L., Chul-Ho J., Lee F., Semi L., Kanamata R., Ji Young K., Mee-Hyun L.e, Hyong Joo L., Ann M B., Ki Won L., Zigang D. USP8 is a novel target for overcoming gefitinib resistance in lung cancer. Clinical Cancer Research. (2013), 14, 3894-3904.

  1. In Fig. 1D, the unit is absent.

Response: Thank you for your recommendation. We have adjusted as suggested.

  1. There are several typographical errors in the main text. The reviewer would request the proofreading.

Response: We would like to thank the reviewer for this suggestion. This has been adjusted as suggested.

Reviewer 2 Report

In the manuscript by Soonnarong et al. the authors have evaluated the effect of artonin F on different lung cancer cell lines. They claim that this compound induces Met ubiquitination and degradation by proteasome, inhibiting PI3K/mTOR/Akt pathway, which induces apoptotic cell death. They also proposed a mechanism that involves palmitoylation and deubiquitanation regulation of Met by this compound. However, most conclusions are not fully supported by the results, being a number of weak points.

Major points:

1-The authors indicate that artonin F induces the ubiquitination and degradation of Met. However, this is not demonstrated. In figure 4 it is shown that upon treatment with the proteasome inhibitor, MG132, an increase in Met levels is produced and a band of a higher molecular weight than 140KDa can also be detected. This band is labeled as “Ub-c-Met” and it is described as Ubiquitin-c-Met. However, according to the described protocols, no antibodies against ubiquitin have been used. Therefore, it has not been demonstrated that this band corresponds to an Ubiquitin form of Met. Moreover, it is known that Met is synthetized as a precursor of 170KDa, which after proteolysis, leads to a fragment of 140KDa and another one of 50KDa, bound by an SH group. As a consequence, the high molecular weight band in the blot in MG132 treated samples could also correspond to the Met precursor (immature form). Therefore, to demonstrate that corresponds to Ubiquitin-c-Met a western-blot against Ubiquitin should be performed.

2-Although the authors indicate that PI3K/mTOR/Akt pathway is inhibited as a consequence of Met degradation, this is not demonstrated. To do it, cells should be treated with HGF to see whether HGF-induced activation of this pathway is decreased. In addition, a Met inhibitor should be used in untreated cells to see whether Met inhibition is able to decrease the levels of phosphorylated mTOR, Akt, etc. On the other hand, in the western-blots shown in figure 2C it is clear that total levels of PI3K, mTOR and Akt decrease when cells are treated with artonin F, in parallel with the decrease in total Met levels. Therefore, their reduced activation (phosphorylated forms) is likely a consequence of a reduction in their levels, not related with their potential activation by Met. This needs to be analyzed using the proteasome inhibitor to see if inhibition of proteasome degradation prevents the reduction in PI3K, mTOR and Akt protein levels. Additionally, an anti-Ubiquitin western-blot should be performed after immunoprecipitation of these proteins.

3-The potential effects of artonin F on enzymes involved in palmitoylation or deubiquitinitation is only based on in silico analysis. It should be demonstrated whether this compound can do it in vitro.

Minor points:

-English should be revised as there are a number of sentences with grammatical mistakes, some missing words, etc.

-In some places of the manuscript Met is described as an oncoprotein, something that is incorrect. Met (or c-Met) is the normal protein encoded by MET protooncogen. In fact, lung cancer cells used in this study overexpress wt Met, not any oncogenic Met protein. This should be corrected.

-The discussion needs to be improved to avoid exhaustive repetition of the results.

-The size of most figures is too small, so it is very hard to see what is written. For example, microscopy images and annexin V graphs from figure 1.

Author Response

Reviewer: 2

In the manuscript by Soonnarong et al. the authors have evaluated the effect of artonin F on different lung cancer cell lines. They claim that this compound induces Met ubiquitination and degradation by proteasome, inhibiting PI3K/mTOR/Akt pathway, which induces apoptotic cell death. They also proposed a mechanism that involves palmitoylation and deubiquitanation regulation of Met by this compound. However, most conclusions are not fully supported by the results, being a number of weak points.

Major points:

1-The authors indicate that artonin F induces the ubiquitination and degradation of Met. However, this is not demonstrated. In figure 4 it is shown that upon treatment with the proteasome inhibitor, MG132, an increase in Met levels is produced and a band of a higher molecular weight than 140KDa can also be detected. This band is labeled as “Ub-c-Met” and it is described as Ubiquitin-c-Met. However, according to the described protocols, no antibodies against ubiquitin have been used. Therefore, it has not been demonstrated that this band corresponds to a Ubiquitin form of Met. Moreover, it is known that Met is synthetized as a precursor of 170KDa, which after proteolysis, leads to a fragment of 140KDa and another one of 50KDa, bound by an SH group. As a consequence, the high molecular weight band in the blot in MG132 treated samples could also correspond to the Met precursor (immature form). Therefore, to demonstrate that corresponds to Ubiquitin-c-Met a western-blot against Ubiquitin should be performed.

Response: We are grateful for the reviewer’s constructive comments, and we have carefully amended the manuscript accordingly. We noticed that we did not indicate about the use of anti-ubiquitin antibody clearly in the first version of the manuscript. We have already revised in method part (Co-immunoprecipitation (Co-IP)), which is further explained in the result section 2.4 (Artonin F decreases c-Met through the induction of c-Met proteasomal degradation) on revised manuscript. In this study, cell lysates are subjected to pull down the c-Met protein by immunoprecipitation and then western blotting with anti-Ubiquitin. Our study revealed that c-Met proteins were degraded by the proteasome, treatment with MG132 inhibitor can be used to supporting this hypothesis. This study indicates that artonin F exerts a promoting effect on c-Met expression by inducing its ubiquitin–proteasomal degradation.

2-Although the authors indicate that PI3K/mTOR/Akt pathway is inhibited as a consequence of Met degradation, this is not demonstrated. To do it, cells should be treated with HGF to see whether HGF-induced activation of this pathway is decreased. In addition, a Met inhibitor should be used in untreated cells to see whether Met inhibition is able to decrease the levels of phosphorylated mTOR, Akt, etc. On the other hand, in the western-blots shown in figure 2C it is clear that total levels of PI3K, mTOR and Akt decrease when cells are treated with artonin F, in parallel with the decrease in total Met levels. Therefore, their reduced activation (phosphorylated forms) is likely a consequence of a reduction in their levels, not related with their potential activation by Met. This needs to be analyzed using the proteasome inhibitor to see if inhibition of proteasome degradation prevents the reduction in PI3K, mTOR and Akt protein levels. Additionally, an anti-Ubiquitin western-blot should be performed after immunoprecipitation of these proteins.

Response: We are grateful for the reviewer’s insightful questions. We found the reviewer’s suggestion about HGF treatment interesting; however, this design of experiment would benefit the compound that inhibit signaling of the c-Met but not the receptor degrading compound.

As this study revealed that artonin F induced c-Met ubiquitin–proteasomal degradation and the receptor for HGF in the treated cells are lost. This is surely that the absence of HGFspecific receptor would be the reason by which of the disruption of down-stream signaling rather than the inhibition of c-Met function.  Regarding the reduction of the phosphorylated from of the proteins as mentioned by the reviewer that may casued by the reduction of the total protiens (PI3K, Akt, mTOR), we re-analysed the ratio of phosphorylated proteins versus their total forms and found that the reductions were depended mainly via the protein signaling (Fig 2D).

Another section of the reviewer's recommendation is analyzed using the proteasome inhibitor to prevent the reduction in PI3K, mTOR, and Akt protein levels. As the PI3K/Akt/m-TOR signaling pathway is widely known to be the down-stream target of c-Met. The reduction of the receptor by artonin F is very likely to cause the reduction of these down-stream protein signals. In addition, we think that the reduction of total protein forms (PI3K, Akt, mTOR) is possibly caused by caspase-dependent activity. In figure 2, the result showed that artonin F at a concentration of 50 µM, the level of cleaved caspase-3 was greatly increased, while the total form of those proteins was reduced.

In addition, we investigated the ability of artonin F to induce c-Met degradation via the ubiquitin-proteasomal pathway, and when the selective proteasome inhibitor (MG132) was used, the level of c-Met-ubiquitin complex in the artonin F-treated cells was significantly increased (Figure 4). Normally, c-Met has a shorter half-life than PI3K, mTOR, and Akt and the experiment of the ubiquitin-c-Met complex showded the increase of unibiquitination of the c-Met as early as 5 h of treatment. In this study, we found that artonin F triggers the degradation of c-Met receptor and decrease the c-Met dowm-stream signals.

3-The potential effects of artonin F on enzymes involved in palmitoylation or deubiquitinitation is only based on in silico analysis. It should be demonstrated whether this compound can do it in vitro.

Response: c-Met has become an attractive target for cancer treatment and drug development. So in this study, we focused on the effect of artonin F on c-Met expression being decreased in cancer cells, the result showed in figure 2 and figure 3. c-Met degradation is a promising therapeutic strategy, it is also the key target in oncological drug development. From immunoprecipitation result, it revealed that artonin F could induce the ubiquitin-c-Met complex. We do agree with the reviewer that the in silico experiment is the additional experiment reveal the possible in detail mechanism of action. As we have focused on the down-regulation of c-Met and decrease of its down-stream signals and we have the wet lab experiment demonstrating it, this specific effect on the palmitoylation or deubiquitination are useful for our further investigation.

In considering the decreased expression of c-Met proteins level in-vitro, we performed an extra-experimental in the in silico section. The results showed that artonin F is likely to act as a c-Met ATP-competitive inhibitor when compared to foretinib (inhibitor targeting MET) the data is provided in result part section 2.7). Altogether, these results could support that artonin F can downregulated c-Met but not operate as a c-Met competitive inhibitor.

  Minor points:

-English should be revised as there are a number of sentences with grammatical mistakes, some missing words, etc.

Response: We would like to thank the reviewer for this suggestion. This has been adjusted as suggested.

-In some places of the manuscript Met is described as an oncoprotein, something that is incorrect. Met (or c-Met) is the normal protein encoded by MET protooncogen. In fact, lung cancer cells used in this study overexpress wt Met, not any oncogenic Met protein. This should be corrected.

Response: Thank you for your suggestion and helpful comments. We have revised this word “Met” to “Met” in discussion part on revised manuscript.

-The discussion needs to be improved to avoid exhaustive repetition of the results.

   Response: Thank you for your recommendation. We have revised to avoid exhaustive repetition of the results in the discussion part.                                                                                                  

-The size of most figures is too small, so it is very hard to see what is written. For example, microscopy images and annexin V graphs from figure 1.

Response: Thank you for your recommendation. We have revised the size of figure1 and figure 3.

Round 2

Reviewer 2 Report

In the revised version of the manuscript by Soonnarong et al. where the authors have evaluated the effect of artonin F on different lung cancer cell lines many conclusions are still not fully supported by the results, remaining a number of weak points.

Major points:

1-As indicated to the authors in the first revision, the decrease in PI3K/mTOR/Akt pathway activation has not been demonstrated to be a consequence of Met degradation. To do it, cells should be treated with HGF to see whether HGF-induced activation of this pathway is decreased. In addition, a Met inhibitor should be used in untreated cells to see whether Met inhibition is able to decrease the levels of phosphorylated mTOR, Akt, etc. None of these experiments have been performed, although necessary to demonstrate it. Moreover, I insist that the western-blots shown in figure 2C indicate that total levels of PI3K, mTOR and Akt decrease when cells are treated with artonin F, in parallel with the decrease in total Met levels. Therefore, their reduced activation (phosphorylated forms) would be a consequence of a reduction in their levels, not related with their potential activation by Met. The authors claim that they have “re-analysed the ratio of phosphorylated proteins versus their total forms and found that the reductions were depended mainly via the protein signaling (Fig 2D). It is difficult to understand the meaning of this sentence. To know if the levels of total PI3K, mTOR or Akt decrease upon treatment with artonin, at least, they should have calculated the ratio between the total level of these protein versus the normalizing protein, GAPDH, whose levels remain unchanged. Moreover, as indicated in the previous evaluation of this manuscript, authors should analyze whether the proteasome inhibitor prevents the reduction in PI3K, mTOR and Akt protein levels. Additionally, an anti-Ubiquitin western-blot should be performed after immunoprecipitation of these proteins. None of these experiments have been done, so that much of the conclusions of the manuscripts have not been demonstrated.

Moreover, the additional explanations given by the authors have no sense: “As the PI3K/Akt/m-TOR signaling pathway is widely known to be the down-stream target of c-Met. The reduction of the receptor by artonin F is very likely to cause the reduction of these down-stream protein signals. In addition, we think that the reduction of total protein forms (PI3K, Akt, mTOR) is possibly caused by caspase-dependent activity. In figure 2, the result showed that artonin F at a concentration of 50 µM, the level of cleaved caspase-3 was greatly increased, while the total form of those proteins was reduced”

Although PI3K/Akt/mTOR pathway is activated by Met, the reduction in total levels of these proteins are not a consequence of the reduction in Met levels. Moreover, phosphorylated levels of these proteins have been determined under normal growing conditions, but not upon stimulation with HGF. Therefore, many other growth factor present in the serum can induce their phosphorylation.

On the other hand, the argument given by the authors that the decrease in PI3K, Akt and mTOR levels could be a consequence of the action of caspases, needs to be demonstrated. I any case, it has not been probed to be a consequence of Met degradation.

2-The potential effects of artonin F on enzymes involved in palmitoylation or deubiquitinitation is only based on in silico analysis. It should be demonstrated whether this compound can do it in vitro.

This remains unanswered by the authors.

  Minor points:

-English should be revised again.

Author Response

RESPONSE TO THE REVIEWER COMMENTS#2

In the revised version of the manuscript by Soonnarong et al. where the authors have evaluated the effect of artonin F on different lung cancer cell lines many conclusions are still not fully supported by the results, remaining a number of weak points.

Major points:

1-As indicated to the authors in the first revision, the decrease in PI3K/mTOR/Akt pathway activation has not been demonstrated to be a consequence of Met degradation.

To do it, cells should be treated with HGF to see whether HGF-induced activation of this pathway is decreased. In addition, a Met inhibitor should be used in untreated cells to see whether Met inhibition is able to decrease the levels of phosphorylated mTOR, Akt, etc. None of these experiments have been performed, although necessary to demonstrate it.

Response :

Thank you very much for these valuable comments that have improved the quality of our works. We have performed several additional experiments and would like to explain our opinions.

As the reviewer would aware that in the absence of HGF the down-stream signals of c-Met would not be activated, however, study has shown that c-Met could be activated via EGFR in the HGF-independent manner (1). There are several studies indicated that the c-Met can interact directly with EGRF and facilitate its down-stream signaling, even independent of HGF (2-4). Crosstalk between c-Met and EGFR in activating down-stream signals controlling cell survival, proliferation, and motility have been evidenced (5, 6). In addition, the HGF was shown to transactivate not only c-Met but also the epidermal growth factor receptor (EGFR) (7), and the activation of EGFR was known to activate the PI3K/AKT/mTOR signal (8).

We performed new experiment to address that artonin F caused the decrease of c-Met receptor as shown in Figure 3. Artonin F reduced the c-Met receptor at early time as 6 h (at high dose), and 12 h (at lower doses). The use of HGF to interact with c-Met and investigated the reduction of down-stream signals may be very good for the mechanism of the tested compound that acts as competitive inhibitor. The competitive inhibitor will competitive with the HGF in binding to the c-Met of the cells. In addition, the system should have the comparable number of c-Met receptor in all groups of the cells (control non-treated group and artonin F treated groups). However, artonin F mechanism of action is not competitive inhibitor, but it induced ubiquitin-proteosomal degradation of the c-Met receptor. So the treatment of HGF in the cells with less number of receptors versus non-treated control may surely cause the reduction of the down-stream signals. However, in the case of c-Met-EGFR crosstalk, using HGF may be difficult to interpreted. We are focusing on the action of compound to decrease the level of receptor. Nevertheless, we think that either HGF-dependent or independent manner, the drug eliminating c-Met receptor should be able to disrupt the down-stream signal of PI3K/AKT/mTOR.

References

  1. A M Dulak, C T Gubish, L P Stabile, C Henry and J M Siegfried. HGF-independent potentiation of EGFR action by c-Met. Oncogene. 2011, 30, 3625–3635.
  2. Shawna Leslie Organ. An overview of the c-MET signaling pathway.  Ther. Adv. Med. Oncol. 2011, 3, S7–S19.
  3. Zhang, Yazhuo et al. Function of the c-Met receptor tyrosine kinase in carcinogenesis and associated therapeutic opportunities. Molecular cancer. 2018. 17, 45.
  4. Giuditta Viticchiè and Patricia A. J. Muller. c-Met and Other Cell Surface Molecules. Biomedicines. 2015,  3, 46-70.
  5. Interaction, Activation and Functional Consequences
  6. Jagadeeswaran R, Puri N, Ma PC, Jagadeesh S, Schaefer E, Christensen J, et al. c-Met/HGF pathway and the role of reactive oxygen species in small cell lung cancer cells. AACR Meeting Abstracts. 2004,42.
  7. Puri N, Ahmed S, Janamanchi V, Treitokova M, Krausz T, Jagadeeswaran R, et al. c-Met is a new therapeutic target for treatment of human melanoma. Clin. Cancer Res. 2007, 7, 2246-53.
  8. Thomas E. Reznik, Yingying Sang, Yongxian Ma, Roger Abounader, Eliot M. Rosen, Shuli Xia, and John Laterra. Transcription-Dependent Epidermal Growth Factor Receptor Activation by Hepatocyte Growth Factor. Mol Cancer Res. 2008,  1, 139–150.
  9. Freudlsperger, C., Burnett, J. R., Friedman, J. A., Kannabiran, V. R., Chen, Z., & Van Waes, C. EGFR-PI3K-AKT-mTOR signaling in head and neck squamous cell carcinomas: attractive targets for molecular-oriented therapy. Expert opinion on therapeutic targets, 2011, 15(1), 63–74.

Moreover, I insist that the western-blots shown in figure 2C indicate that total levels of PI3K, mTOR and Akt decrease when cells are treated with artonin F, in parallel with the decrease in total Met levels. Therefore, their reduced activation (phosphorylated forms) would be a consequence of a reduction in their levels, not related with their potential activation by Met. The authors claim that they have “re-analysed the ratio of phosphorylated proteins versus their total forms and found that the reductions were depended mainly via the protein signaling (Fig 2D). It is difficult to understand the meaning of this sentence. To know if the levels of total PI3K, mTOR or Akt decrease upon treatment with artonin, at least, they should have calculated the ratio between the total level of these protein versus the normalizing protein, GAPDH, whose levels remain unchanged. Moreover, as indicated in the previous evaluation of this manuscript, authors should analyze whether the proteasome inhibitor prevents the reduction in PI3K, mTOR and Akt protein levels. Additionally, an anti-Ubiquitin western-blot should be performed after immunoprecipitation of these proteins. None of these experiments have been done, so that much of the conclusions of the manuscripts have not been demonstrated.

Moreover, the additional explanations given by the authors have no sense: “As the PI3K/Akt/m-TOR signaling pathway is widely known to be the down-stream target of c-Met. The reduction of the receptor by artonin F is very likely to cause the reduction of these down-stream protein signals. In addition, we think that the reduction of total protein forms (PI3K, Akt, mTOR) is possibly caused by caspase-dependent activity. In figure 2, the result showed that artonin F at a concentration of 50 µM, the level of cleaved caspase-3 was greatly increased, while the total form of those proteins was reduced”

Although PI3K/Akt/mTOR pathway is activated by Met, the reduction in total levels of these proteins are not a consequence of the reduction in Met levels. Moreover, phosphorylated levels of these proteins have been determined under normal growing conditions, but not upon stimulation with HGF. Therefore, many other growth factor present in the serum can induce their phosphorylation.

On the other hand, the argument given by the authors that the decrease in PI3K, Akt and mTOR levels could be a consequence of the action of caspases, needs to be demonstrated. I any case, it has not been probed to be a consequence of Met degradation.

RESPONSE: For these comments, the reviewer is right about the additional experiments we should perform. In addition, the experiment demonstrating that the total proteins were reduced by caspase-dependent mechanism should be added.

We have performed the new experitments demonstrating the specific decrease of c-Met in response to artonin F at early time points and the caspase inhibitor could prevent the reduce of total PI3K, AKT, and mTOR proteins. There are 3 new experiments added into the revised manuscript.

 1) The immunofluorescence analysis shows that the artonin F caused the decrease of c-Met receptor in the lung cancer cells as early as 6 h after treatment at the concentration of 50 µM (Figure 3A in the revised Manuscript). At the concentrations of 20-50 µM, artonin F reduced the c-Met as early as 12 h.

2) the western blot analysis at various time 6-24 h, showing the specific reduction of c-Met at early time but not other proteins (PI3K, AKT, and m-TOR). These results support the specific effect of artonin F on c-Met.

3) We used the caspase inhibitor Z-VAD-FMK to inhibit the caspase activity, as we have hypothesized that the total forms of other proteins were decreased in response to aftonin F treatment at 24 h as a result from caspase activity. We found that the caspase inhibitor could dramatically prevent the decrease of such proteins in the present of artonin F (Figure 2E, F).

Regarding the way of normalization, as we have demonstrated that the decrease of total forms of PI3K, AKT, and mTOR was a result of caspase activation, we wished to remain normalizing the phosphorylated proteins with their total forms.

2-The potential effects of artonin F on enzymes involved in palmitoylation or deubiquitinitation is only based on in silico analysis. It should be demonstrated whether this compound can do it in vitro.

RESPONSE: We do agree with the reviewer that the in silico experiment is an additional experiment that reveals the possible in-depth mechanism of action.

However, there are several good works have reported the in dept mechanisms using in silico without the wet lab experiments as followed. Previous research has demonstrated that the in silico method is also useful in terms of determining the target or potential of a natural cytotoxic chemical (1, 2). Previous investigation, for example, investigated in silico for a novel human USP2 inhibitor to inhibit the SARS-CoV-2 papain-like (PLpro) protease that is utilized to treat COVID-19 (3). Furthermore, in silico research was performed to examine the mechanism and function of deubiquitination (4), as well as to predict the site of S-Palmitoylation synaptic proteins in rat (5). Moreover, in silico method was employed to conduct an in-depth analysis of the P2X7 cytoplasmic domain receptor in order to establish the process of palmitoylation in order to prevent channel desensitization (6). However, previous studies showed that Imidazopyridine hydrazone derivatives exert antiproliferative effect on lung and pancreatic cancer cells and potentially inhibit receptor tyrosine kinases including c-Met. The result showed molecular docking and dynamics simulation studies corroborated the experimental findings and revealed possible binding modes of the select derivatives with target receptor tyrosine kinases (7).  As a result, studying the activity of artonin F while utilizing in silico techniques to estimate the substance's potential through these processes is sufficient to address the fundamental in-depth concerns.

In closing, we do agree that the wet lab experiments confirming the in silico results would be ideal. However, as we have focused on the effect of the compound to decrease the c-Met receptor and we have shown the wet lab results in several figures. We may request for the opportunity to publish our work in this recent form.

References

  1. S. Dutta G., and P. K., In Silico Target Identification and Molecular Docking Studies of Natural Cytotoxic Compound Borivilianoside H. Curr. Biotechnol. 2019, 8, 127-137.
  2. Seshu V. and Suban K.S., In silico ADMET and molecular docking study on searching potential inhibitors from limonoids and triterpenoids for COVID-19. Comput. Biol. Med. 2020, 124, 1-12.
  1. Muhammad U.M., Sarfraz A., Iskandar A. and Matheus F. Identification of novel human USP2 inhibitor and its putative role in treatment of COVID-19 by inhibiting SARS-CoV-2 papain-like (PLpro) protease. Comput. Biol. Chem. 2020, 89, 1-8.
  2. Rashmi K. Shrestha, Judith A. Ronau, Christopher W. Davies, Robert G. Guenette, Eric R. Strieter, Lake N. Paul, and Chittaranjan Das. Insights into the Mechanism of Deubiquitination by JAMM Deubiquitinases from Cocrystal Structures of the Enzyme with the Substrate and Product. Biochemistry. 2014, 53, 3199-3217.
  3. Soumyendu S. B., Anup K. H., Monika Zareba-K., Anna Bartkowiak-K., Aviinandaan D., Piyali C., Mita N., Tomasz W., Jakub W., and Subhadip B. RFCM-PALM: In-Silico Prediction of S-Palmitoylation Sites in the Synaptic Proteins for Male/Female Mouse Data. Int. J. Mol. Sci. 2021. 22. 1-14.
  4. Alanna E.McC., Craig Y., Steven E.M., Full-Length P2X7 Structures Reveal How Palmitoylation Prevents Channel Desensitization. Cell. 2019, 179, 659-670.
  5. Tahereh D., Fatemeh M., Mehdi K., Motahareh M., Somayeh P., Zahra K., Luciano S., Najmeh E. and Omidreza F. Imidazopyridine hydrazone derivatives exert antiproliferative effect on lung and pancreatic cancer cells and potentially inhibit receptor tyrosine kinases including c-Met. Scientific Reports. 2021, 11, 1-15.

  Minor points:

-English should be revised again.

RESPONSE: We have sent the English correction for the second round by Enago to ensure the quality of the English. Please find the attached Certificate of English Editing.

Round 3

Reviewer 2 Report

In this new revised version of the manuscript by Soonnarong et al. where the authors have evaluated the effect of artonin F on different lung cancer cell lines some conclusions are still not fully supported by the results, remaining a number of unanswered weak points.

Major points:

1-As indicated to the authors concerning previous versions of the manuscript, the decrease in PI3K/mTOR/Akt pathway activation induced by artonin has not been demonstrated to be Met dependent, although in several places along the manuscript (including the abstract) the authors say that. In order to demonstrate it, at least, a Met inhibitor should be used in untreated cells to see whether Met inhibition is able to decrease the levels of phosphorylated mTOR, Akt, etc. However, this experiment has not been performed yet, although necessary to demonstrate it. In the answer to this reviewer, the authors claim that Met can be activated in the absence of HGF, or even they discuss the possibility of its cross-talk with EGFR, which is true, but in the manuscript they still say that PI3K signaling decrease depends on MET, which has not been demonstrated in this context. In any case, they are Met inhibitors that can be used to test if they are able to inhibit the decrease in PI3K, Akt and mTOR levels upon treatment with artonin.  

2-On the other hand, and in relation to the first point, the authors now show in the new figure 3 that total levels of PI3K, AKT and mTOR upon treatment with artonin plus Z-VAD do not decrease. However, in this experiment the effect of artonin alone is missing. It is always necessary to do the full experiment with all the conditions at the same time. In addition, it is absolutely necessary to calculate the ratio between total PI3K/GADPH, Akt/GADPH and mTOR/GADPH levels. It looks that mTOR/GADPH levels decrease upon treatment with artonin (20 and 50 mM) at 12h.

2-The results shown in figure 5 lack of some missing controls: untreated cells and artonin treated cells in the absence of MG132.

3-The potential effects of artonin F on enzymes involved in palmitoylation or deubiquitinitation is only based on in silico analysis. It should be demonstrated whether this compound can do it in vitro.

4-All the manuscript should be double checked in order to avoid the inclusion of sentences  and conclusions non based on results.

Minor points:

-English should be revised again, as well as size of figures as they are too small.

Author Response

I provide response according to the latest Editor suggestion which highlighted in green color as shown below.

Some of the responses by the authors to comments by reviewer 2 (in review 3) are appropriate (as noted below). However, there are a few small issues that the authors could easily address. We suggest that the authors specifically respond to the following points that were raised by reviewer 2 (in review 3):

In response to the comment 1, the authors have made a reasonable argument. However, they need to alter the language they use to NOT state that PI3K/Akt/mTOR changes are dependent on Met to fully address the issue raised by the Reviewer.

Response We would like to thank the editor for this suggestion. According to the Editor’s suggestion, we have modified the sentence of “artonin F inhibit c-Met dependent PI3K signaling” to “artonin F decreased the levels of c-Met and PI3K” in the part of result 2.3 in revised manuscript. In discussion part, we have modified the sentence of “artonin F mediated apoptosis by inhibiting the c-Met dependent PI3K/Akt/mTOR” to “artonin F mediated apoptosis by downregulating of c-Met and decreasing PI3K/Akt/mTOR signaling”.

Concerning comment 2, the technical control of artonin alone is missing. But given the data presented previously in the manuscript, the authors do not need to address it.

Response We would like to thank the editor for this point. We greatly appreciate your kindness.

Regarding comment 2 on calculating the ratio between total PI3K/GADPH, Akt/GADPH and mTOR/GADPH levels, it is not clear why have authors not done this. It is easy to do. It certainly appears that total protein levels are decreased with high dose Artonin F in Fig. 2. In Fig 3 – it becomes apparent this total protein loss is at 24h, but it is less clear at earlier time point. The ratios of total protein to GAPDH should be provided.

Response Thank you very much for raising this point, we have modified the Figure 2 and Figure 3 in revised manuscript. We modified the graph to calculate the total protein compared to GAPDH and the phosphor-protein when compared to GAPDH to make it clearer and easier to understand.

Regarding comment for Fig. 5, the controls mentioned by the reviewer are missing – but will not change result. So, the authors do not need to address this.

Response We would like to express our gratitude for your assistance.

Regarding the comment “-The potential effects of artonin F on enzymes involved in palmitoylation or deubiquitinitation is only based on in silico analysis. It should be demonstrated whether this compound can do it in vitro”, it is our opinion that a response to this is not needed. Ok as is as a preliminary further dive into mech. We suggest that the authors modifiy language to indicate the prelimary and suggestive in silico nature of this data.

Response Thank you very much for raising this point. We included this statement in the results and discussion part. And to avoid the reader’s misunderstanding, we used preliminary studty in silico models to predict mechanism of action of artonin F, which downregulated c-Met.

Regarding the need to improve the language of the manuscript, we agree with the reviewer’s comments, as indicated below:

Authors really need to be more careful with language and seem to be mis-stating results. A few examples are:

  1. cMET not necessarily “inhibited” (title for Fig 2 section) – its abundance is
  2. Title: AKT signaling is not “inhibited” – it is

Response We are grateful for the editor suggestion, and we have carefully amended the manuscript accordingly.

  • Title: Artonin F induces ubiquitin-proteasomal degradation of c-Met and decreases Akt-
    mTOR signaling
  • Abstract: we report the downregulation of c-Met by artonin F
  • Title for result 2.2: Artonin F triggers apoptosis cascade through mechanisms that involve
    c-Met downregulation
  • Result 2.2: Thus, the mechanism of action of artonin F can reduce c-Met, as well as PI3K,
    Akt and mTOR.
  • Title for figure 2: Artonin F triggers the apoptosis through decreased c-Met and
                                     PI3K/Akt/mTOR signaling in lung cancer cells.
  • Title for result 2.3: Artonin F decreased the levels of c-Met and p-PI3K
  • Result 2.3: artonin F decreased the levels of c-Met and p-PI3K

  These findings suggest that artonin F has a c-Met downregulating effect

  • Title for figure 4: Artonin F decreases the levels of c-Met and p-PI3K.
  • Discussion: artonin F mediated apoptosis by downregulating of c-Met and decreasing
    PI3K/Akt/mTOR signaling

                     - we provided for artonin F mediated c-Met downregulation and 
                        PI3K
downstream signal transduction

                    - Artonin F exhibits potential downregulation of c-Met by
                       enhancing the ubiqui
tin-proteasomal degradation of c-Met.
